# Computational Detection of Breast Cancer Invasiveness with DNA Methylation Biomarkers

**DOI:** 10.3390/cells9020326

**Published:** 2020-01-30

**Authors:** Chunyu Wang, Ning Zhao, Linlin Yuan, Xiaoyan Liu

**Affiliations:** 1School of Computer Science and Technology, Harbin Institute of Technology, Harbin 150080, China; 2School of Life Science and Technology, Harbin Institute of Technology, Harbin 150080, China; zhaoning2016@hit.edu.cn; 3College of Intelligence and Computing, Tianjin University, Tianjin 300350, China; yuan_linlin@163.com

**Keywords:** breast cancer, metastasis, invasiveness, DNA methylation

## Abstract

Breast cancer is the most common female malignancy. It has high mortality, primarily due to metastasis and recurrence. Patients with invasive and noninvasive breast cancer require different treatments, so there is an urgent need for predictive tools to guide clinical decision making and avoid overtreatment of noninvasive breast cancer and undertreatment of invasive cases. Here, we divided the sample set based on the genome-wide methylation distance to make full use of metastatic cancer data. Specifically, we implemented two differential methylation analysis methods to identify specific CpG sites. After effective dimensionality reduction, we constructed a methylation-based classifier using the Random Forest algorithm to categorize the primary breast cancer. We took advantage of breast cancer (BRCA) HM450 DNA methylation data and accompanying clinical data from The Cancer Genome Atlas (TCGA) database to validate the performance of the classifier. Overall, this study demonstrates DNA methylation as a potential biomarker to predict breast tumor invasiveness and as a possible parameter that could be included in the studies aiming to predict breast cancer aggressiveness. However, more comparative studies are needed to assess its usability in the clinic. Towards this, we developed a website based on these algorithms to facilitate its use in studies and predictions of breast cancer invasiveness.

## 1. Introduction

According to a National Cancer Center report describing the status and trends of cancer in China in 2017, breast cancer is the most common female malignancy in the country. It is a complex and heterogeneous disease with multiple molecular subtypes, which can be defined by immunohistochemistry or microarray profiling [1,2,3]. The incidence of breast cancer is increasing, but there are limited curative options when metastasis develops. Mammography has been shown to be an economical non-invasive tool for early diagnosis [4]. However, the high mortality rate of breast cancer is, to a large extent, a result of metastasis, which affects up to 40% of women suffering from this disease [5]. Unfortunately, despite the continuous improvements of medical technology, we still cannot control cancer metastasis[6].

Metastasis refers to the process in which malignant tumor cells relocate and then continue to grow in other parts of the body separate from the primary site, by traveling through lymphatic channels, blood vessels, or the body cavity; it is often the main reason for the failure of tumor treatment [7,8,9,10,11,12]. The occurrence and progress of metastases in tumors are nonrandom and thus potentially predictable. For example, colorectal cancer typically spreads to the liver, while breast cancer primarily metastasizes to bone marrow and lung [13]. This emphasizes the need for novel prognostic tools to guide clinical decision-making about diagnosis and treatment. Predicting the invasiveness and progression of tumors is crucial for clinical decision-making and avoiding both overtreatment of indolent breast cancer and undertreatment of aggressive disease [14].

In view of this, several groups have analyzed the molecular expression profiles of primary tumors and metastases to regional lymph nodes or distant sites [15,16,17,18,19]. In recent years, large-scale sequencing techniques, such as next-generation sequencing and microarray [20,21,22,23], have enabled the systematic detection of abnormalities of the genome, transcriptome, and epigenome associated with cancer [22,24,25,26,27,28,29,30,31,32,33,34]. One can reevaluate the cancer progress through an integrative way [35] to understand their regulation system [36,37]. Moreover, this has been enhanced by single-cell level omics [38]. However, metastatic cancer-related data are still extremely scarce, which is a result of the complexity of metastasis. The recurrence and metastasis of tumors usually occur several years after the primary tumor has been removed [39]. To investigate the possibility of metastasis of a primary tumor occurring long after treatment, follow-up should be performed for a long time, which is expensive and laborious [40]. Moreover, as secondary cancer growing at new sites cannot be routinely removed like the primary tumor during treatment for metastatic breast cancer, the tissue is not available for research until an autopsy is performed [41].

Cancer-specific alterations of DNA methylation are closely related to a variety of malignancies [42,43,44,45]. Recent studies investigating the genomic lesions of primary and metastatic cancers revealed that some specific DNA methylation changes could account for tumor metastasis and progression [46,47]. The appearance of lymph node metastasis is an indication of tumor cells developing the ability to leave the primary site and spread to a new site; it thus acts as a marker of the ability of the tumor to establish distant metastases [48].

In this study, we used DNA methylation data for lymph node metastasis to study the invasiveness of breast cancer. To overcome the problems of a small amount of data and lack of samples for metastatic cases, we used a novel method to identify sample labels based on their DNA methylation markers and then constructed a classifier for identifying invasive breast cancer. Upon applying this classifier to The Cancer Genome Atlas (TCGA) BRCA samples, the acquired results were satisfactory.

## 2. Materials and Methods

Figure 1 provides an overview of the experimental procedure. It consists of four major components: (i) differential methylation analysis selecting specific CpG sites, (ii) filtering of redundant feature sites by dimensionality reduction, (iii) building classifiers based on the Random Forest algorithm, and (iv) evaluation of classifier performance by enrichment analysis.

### 2.1. Materials: DNA Methylation Datasets

In this study, we used three genome-wide DNA methylation profiles of breast cancer. The first two datasets were acquired from the Gene Expression Omnibus (GEO). One of them includes 44 matched primary breast cancer and regional metastases (GSE58999) [49], while the other contains 80 primary breast cancer and 40 normal samples (GSE66695) [50]. Another dataset was obtained from The Cancer Genome Atlas (TCGA); this dataset comprises data on a primary breast cancer population (n=766) and a normal population (n=97). The DNA methylation profile was measured by Infinium HumanMethylation450 BeadChip (HM450), which contains more than 480,000 probes. The HM450 DNA methylation profile was also used, which covers 99% of the NCBI reference genes and can provide data on DNA methylation at single-base resolution [51].

In the pretreatment processing, we deleted single-nucleotide polymorphism (SNP) probes and CpH (A/T/C) sites and then removed CpG sites at which more than 30% of samples had missing values (NA). The remaining missing values were complemented by the k-nearest neighbor, using the “impute.knnl” function in the R package “limma.”

### 2.2. Methods: Study Design

We calculated Euclidean distances between any two matched samples using all 393,806 probes, and then removed abnormal sample pairs with excessive distances between them using the method of quartile. Subsequently, we obtained 40 matched primary breast cancer and lymph node metastatic sample pairs whose clinical information was obtained from GSE58999 and in which primary breast cancer samples were defined as invasive. Furthermore, in order to determine whether the primary breast cancer samples in GSE66695 are invasive, we calculated Euclidean distances between lymph node metastatic samples and these primary breast cancer samples using all probes. We believe that the greater the distance between the sample and the lymph node metastatic samples the more likely it is to be noninvasive. Therefore, if the minimum distance between a sample and the 40 lymph node metastases was 10 units larger than the maximum distance of the paired sample, we defined the sample as noninvasive. In this way, we identified 20 noninvasive samples. Invasive (40 samples) and noninvasive (20 samples) samples, together with 40 normal samples, collectively constituted the training sets.

#### 2.2.1. Robust Empirical Bayes

Empirical Bayes (EB) is a statistical algorithm that assumes a Bayesian hierarchical model for the variances and estimates the prior distribution from the marginal distribution of the observed data [52]. Robust EB improves differential methylation analysis by strengthening the hyperparameter estimation procedure and achieves robustness with regard to inaccurate working priors by conditioning on the rank of each estimated log-fold change rather than on the actual observation. It uses log-expression values to fit linear models for each CpG site; computes its moderated t-statistic, moderated F-statistic, and log-odds of differential methylation; and selects differential methylation sites by implementing robust hyperparameter estimation [53]. This method was applied using the R package “limma” [54].

#### 2.2.2. Significance Analysis of Microarrays

Significance Analysis of Microarrays (SAM) establishes a d-statistic for each site, correlating these sites with an outcome variable, such as metastasis, after which CpG sites are rearranged in descending order of d-statistic. SAM simulates a null distribution by permutating the mark of the group randomly to calculate the p-value of the difference in methylation between groups [55]. The permutation algorithm rearranges and repartitions samples. This process is performed M times, and for site i, its statistic of pth permutation is recorded as $d_{p}\left(i\right)$. The sites are rearranged in descending order of $d_{p}\left(i\right)$. Note that:(1)$$\begin{array}{c} d_{E}\left(i\right)=\underset{p}{\sum }\frac{d_{p}\left(i\right)}{M} \end{array}$$
(2)$$\begin{array}{c} \Delta =\left|d\left(i\right)-d_{E}\left(i\right)\right| \end{array}$$

Here, screening is performed for differential methylation sites as those with Δ greater than the threshold. This method was carried out using the R package “samr” [56].

#### 2.2.3. Maximum Relevance Maximum Distance

Maximum Relevance Maximum Distance (MRMD) is a Java-based feature selection method [57]. It aims to select features with maximum relevance and maximum distance. It uses Pearson’s correlation coefficient to measure the correlation of feature and label, and uses Euclidean distance between features to calculate redundancy, which was also widely used in clustering [58,59,60]. MRMD ranks all candidate features based on the calculated Pearson’s correlation coefficient and Euclidean distance, then constructs a simple classifier using the top-ranked features, and finally selects a feature list with the best classification accuracy. MRMD can pick out the optimal number of features, having the lowest redundancy and the strongest correlation with the categorical variable.

#### 2.2.4. Minimal Redundancy Maximal Relevance

Minimum Redundancy Maximum Relevance (mRMR) is a filtered feature selection method [61]. mRMR is based on the concept of minimizing the correlation between different features, while maximizing the correlation between features and target classes. Each correlation is measured based on mutual information. Such mutual information can be regarded as the amount of information about another random variable contained in a random variable, which is a measure of the statistical correlation between two random variables [62]. After using mRMR for feature selection, the ranking of each feature regarding its importance is obtained. Next, cross-validation is performed to select the subset of features with the best performance [63,64,65,66].

#### 2.2.5. Principal Component Analysis

Principal Component Analysis (PCA) is one of the most widely used data compression algorithms. It maps n-dimensional data to k-dimensional space by linear projection (k<n) and obtains the new data with the largest variance in the projected dimension; it results in fewer data dimensions being used and more features of the original data being retained [67]. The principal components are selected based on the cumulative percentage of total variation. We chose to retain the number of principal components representing more than 95% of the total variation, which is a frequently used threshold [68]. We implemented PCA using the Dimensionality Reduction feature of scikit-learn [69].

#### 2.2.6. Factor Analysis

Factor Analysis (FA) is a statistical technique of extracting common factors from a variable population [70]. The common factor refers to the inherent hidden factor between different variables. FA is based on the concept of classifying the observed variables; highly correlated variables are grouped into the same class, while variables in different categories are poorly correlated. Each type of variable actually represents a basic structure, a common factor. In this way, most of the information of the original data can be reflected by a few factors, which enables the data to be condensed. FA was carried out using the Dimensionality Reduction part of scikit-learn [69].

#### 2.2.7. Unsupervised Hierarchical Clustering

Heatmaps were used to display the difference of DNA methylation levels between invasive and noninvasive groups. We randomly chose 10 samples from each of the two patient groups and applied the “levelplot” function implemented in the R package “lattice” to visualize their difference [71]. The unsupervised hierarchical clustering was performed with the hclust function in R (method = “complete”).

#### 2.2.8. Constructing the DNA Methylation-Based Invasiveness Classifier

After differential methylation analysis and dimensionality reduction, we generated a list of 134 variably methylated CpG sites between the invasive and noninvasive groups and a list of 14 variably methylated CpG sites between normal and cancer groups. We used these sites as features to construct classifiers. The Random Forest model is a successful ensemble learning classifier. It can build a series of classification and decision trees through training and has been proven to perform well for the classification of multi-gene microarray chip data [55]. We therefore built classifiers based on a random forest algorithm and used 10-fold cross-validation to evaluate the accuracy of the model.

#### 2.2.9. Hypergeometric Test

Hypergeometric tests were used to determine whether sample groups were particularly associated with some clinical factors. The use of a hypergeometric distribution is a common method of enrichment analysis [72]. It calculated the probability of enrichment of a clinical factor for each class of samples, taking the enrichment ratio of a whole sample set in this clinical factor as the background. The p-values of multiple-test correction were adjusted using the FDR method.

## 3. Results

### 3.1. Feature Selection

Differential methylation analyses were performed using the methods of Robust EB and SAM. For the results of these two methods, we set thresholds of logFC≥1.5 (fold change) and p-value <0.01 to select differentially methylated sites. Robust EB identified 8653 and 11,808 CpG sites in the cancer–normal group and invasiveness–noninvasiveness group, respectively. SAM selected 14,096 and 7329 CpG sites in the cancer–normal group and invasiveness–noninvasiveness group, respectively. The false discovery rate (FDR) of both methods was set to 0.01. To further reduce the false positive rate, we selected the results that overlapped between Robust EB and SAM. Using the overlapping results, there were 7888 and 6461 sites that were differentially methylated between the two groups (the detailed information on CpGs is available in the Appendix A).

In the process of tumor occurrence and development, methylation alert occurs in a genome-wide scale, and many changes are consistent. Therefore, there are many redundant features. In this context, further dimensionality reduction is required to filter out redundant CpG sites. Here, we used four dimensionality reduction methods to select the optimal number of features, namely, MRMD, mRMR, PCA, and FA. MRMD and mRMR are feature selection methods, which identify CpG sites that are related to tags but not related to each other. PCA and FA combine the original features into new features to achieve dimensionality reduction and lose as little of the information conveyed in the original data as possible. To compare the effects of the different dimensionality reduction methods, we constructed classifiers with the four sets of associated results. The dimensionality reduction results and classifier accuracy of the four methods are listed in Table 1.

### 3.2. Development of Classifier Based on DNA Methylation Biomarkers

Next, we established a DNA methylation-based BRCA invasiveness classifier to categorize primary breast cancer as either invasive or noninvasive using the four dimensionality reduction results. We used the Random Forest algorithm to train the classifier and confirmed the validity of the classification by 10-fold cross-validation. With regard to training accuracy, the four results were all satisfactory. The accuracy of the normal prediction was as high as 99%, and the prediction accuracy of invasion was as high as 95%. The classification results are listed in detail in Table 1. To provide more useful information, we also added a false positive (FP) and false negative (FN) in the Appendix A (Appendix A). However, the performance of the classifier is more reflected in the test accuracy. For this reason, we downloaded an independent dataset from TCGA database to test the classifiers.

### 3.3. TCGA Beast Cancer Cohort Confirms the Performance of Classifiers

To test the classifier on a new dataset, we used the publicly available breast cancer (BRCA) HM450 DNA methylation data and accompanying clinical data from The Cancer Genome Atlas (TCGA) project. We tested 766 primary breast cancer samples and 97 normal samples using the classifiers. The accuracy of the four classifiers for normal samples was as high as 96.9%, almost as high as that for the training. However, for the prediction of invasiveness, the four dimensionality reduction methods provided markedly different results.

To evaluate the predictive performance of the classifier, we investigated some metastasis-related clinical indicators of samples from the two groups of prediction. We applied the hypergeometric test to confirm the significance of the enrichment of prediction samples in some clinical indicators, such as primary tumor stage (T-stage), regional lymph node stage (N-stage), human epidermal growth factor receptor 2 (HER2) status, and presence of lymph node metastasis (LN+) in pathological report. T3 indicates that the maximum diameter of the primary breast tumor is more than 5 cm, while N3 refers to lymph node metastases of ipsilateral internal mammary and ipsilateral. HER2 positivity (HER2+) suggests that breast cancer is susceptible to relapse or metastases. LN+ indicates that lymph node metastases are mentioned in the pathological report. These indicators all reflect greater invasiveness [73]. According to the prediction labels of the four classifiers, we applied the hypergeometric test on these indicators separately. Figure 2 shows comparisons of the enrichment ratio and significance of the four results. The histogram above the X axis represents the enrichment ratio of the sample population on the clinical feature, which is greater than 1. We expect to see more enrichment of the sample predicted to be invasive on clinical factors suggesting metastasis. In terms of the dimensionality reduction results of MRMD, samples predicted to be invasive were significantly enriched for indicators suggesting susceptibility to tumor metastasis, such as T3, N3, and HER2+. The detection of LN+ is associated with the number of examined lymph nodes, and the number of examined lymph nodes is related to the scope of the lymph node dissection during the operation. There are some other pathways to metastasize, such as direct infiltration and blood transfer [74]. These factors probably lead to inaccuracy of measuring invasiveness with LN+ and account for the poor significance of IN+. Two indicators were extremely significant under the effect of PCA, and two were significant in the result of FA. These results all demonstrate that our classifier is effective.

### 3.4. Methylation Differences between the Invasive and Noninvasive Groups

Alterations in DNA methylation occur in all cancers, play important roles in the development and progression of cancer, and are associated with tumor aggressiveness in most types of cancer [42]. In this study, we identified noninvasive primary tumor samples by the degree of dissimilarity of DNA methylation between primary cancer and lymph node metastases. Next, we applied two statistical methods to search for probes that were differentially methylated between the invasive and noninvasive groups. Based on the overlapping results between robust EB and SAM, 6461 CpG sites were revealed. An excessive number of features would increase the complexity of the model and lead to overfitting. We therefore further reduced the dimensions to filter out redundant CpG sites and used the optimal number of features to build classifiers. Finally, we evaluated the performance of these classifiers through enrichment analysis. In terms of the biological analysis results, MRMD had the best dimensionality reduction effect, which retained 134 differentially methylated sites (Appendix A). Therefore, we further explored these 134 CpG sites.

To investigate whether this method of sample division and feature selection can effectively identify CpG sites associated with metastasis, we performed unsupervised hierarchical clustering on 20 randomly selected samples based on their methylation levels on these 134 sites. The heatmap shows the results of unsupervised clustering and the difference of methylation patterns between the two types of sample. As shown in Figure 3, the two samples were completely separated and the methylation levels also showed significant differences.

### 3.5. Genes Related to Metastasis

We further listed the genes that the 134 CpG sites correspond to, giving a total of 98 (Appendix A). The HCMDB database annotated most of these genes as being associated with metastasis [75]. Researchers systematically reviewed more than 7000 published papers in the PubMed database using “metastasis” and corresponding gene symbols as the keywords, and manually curated 2183 genes related to metastasis. Comparing the 98 genes with these metastatic cancer-related genes confirmed in the literature, we found that 12 genes were associated with metastatic cancer, which were confirmed in the literature, among which five were associated with breast cancer metastasis. Table 2 lists the 12 confirmed metastasis-related genes and associated literature annotations, with black indicating the genes related to breast cancer metastasis. This indicates that our method can effectively detect CpG methylation sites related to cancer metastasis and also provides new information for the discovery of more relevant genes.

Taking into account the dimensional reduction results of MRMR, only five sites are needed to completely separate the training set. We also analyzed the corresponding genes for these five sites. The results show that two of the sites are on the gene body, while the remaining three are currently not annotated. We suspect that these five sites could become biomarkers suggestive of breast cancer metastasis. In view of this, we have listed information on these five sites in the Appendix A (Appendix A).

### 3.6. Website BMMP

To facilitate research on breast cancer metastasis, we developed a website for the prediction of invasiveness of breast cancer, BMMP (BRCA methylation metastasis prediction [76]). BMMP is a Java-based website that uses the classification model of MRMD’s dimensionality reduction results. BMMP enables users to predict the invasiveness of breast cancer by pasting or uploading DNA methylation profiling data. It also lists experimentally validated metastasis-associated genes used by the classifier. The data for each step of the experiment and classification model are accessible on the website.

## 4. Discussion and Conclusions

Invasive and noninvasive breast cancers differ markedly in their clinical manifestations and prognosis. They also have different clinical treatments. Accurately identifying the differences and predicting tumor invasiveness can have a radical impact on breast cancer research. As an important epigenetic regulator, DNA methylation can serve as a stable marker for samples. In the study reported here, we inferred whether tumors are invasive or noninvasive based on the DNA methylation pattern of breast cancer.

We used two differential methylation analysis methods to identify the CpG sites differentially methylated between the two groups. After reducing the dimensions of these features, we constructed a methylation-based invasiveness classifier to categorize primary breast cancer as either invasive or noninvasive. Finally, we confirmed the credibility of the classifier by comparing the extent to which some clinical factors were particularly enriched in the predicted samples. Our study provides molecular-based support for determining the invasiveness of breast cancer and indicates the potential impact of applying this approach to clinical decision-making. Although this method was only used to evaluate the invasiveness of breast cancer in this study, we believe it should be generally applicable to other types of tumor.

Although this method can guide the study of breast cancer metastasis, it has some limitations. For example, breast cancer is a highly heterogeneous disease, so a fixed classifier may have different prediction accuracies for samples from different subclasses. Considering this, we chose samples including four disease subtypes as a training set [49]. As the study of heterogeneity deepens, research on metastatic cancer has begun to consider this issue, leading to the development of a personalized committee classifier [77], for example. We expect that more intense research can further contribute to revealing the molecular mechanisms involved in breast cancer metastasis and potentially help in diagnosing and treating it. Furthermore, link prediction [78,79,80,81,82], probabilistic models [83,84,85,86] and computational intelligence methods [20,87,88,89,90], which have been successfully applied in many areas [63,91,92,93,94,95,96,97,98,99,100], can be considered in BRCA methylation metastasis prediction.

## Figures and Tables

**Figure 1 cells-09-00326-f001:**
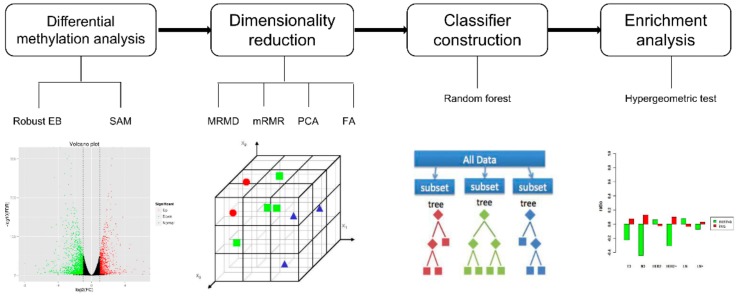
Schematic diagram of the presented stepwise analysis. Step 1, differential methylation analysis: Two methods were carried out to select the differential methylation sites. Step 2, dimensionality reduction: using four methods to reduce the dimension. Step 3, four kinds of dimensionality reduction results were used to construct the classifier respectively. Step 4, enrichment analysis: Hypergeometric test evaluated and compared the performances of the classifiers.

**Figure 2 cells-09-00326-f002:**
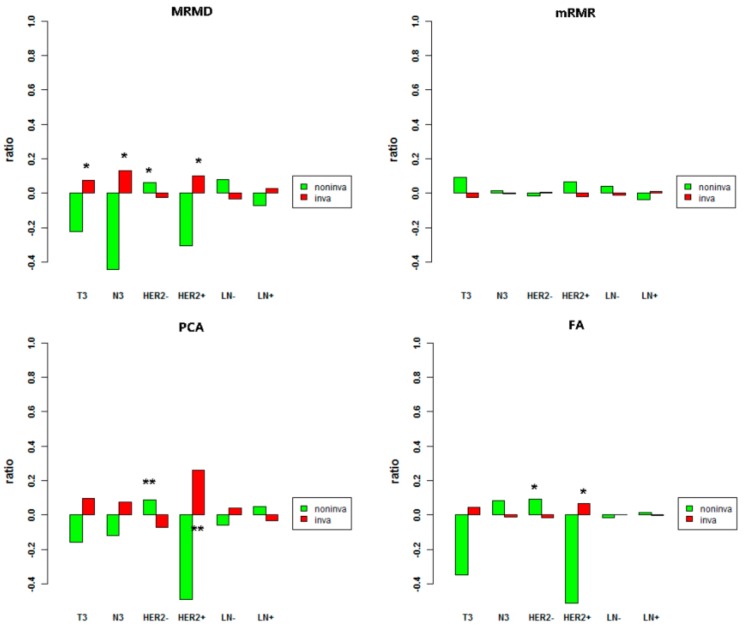
Four enrichment analyses results of clinical indicators in the two tumor clusters. Red indicates samples predicted by the classifier as invasive, and green indicates samples predicted to be non-invasive. The X axis represents clinical indicators; the Y axis represents the enrichment ratio of Table 0. ** is extremely significant (p<0.01).

**Figure 3 cells-09-00326-f003:**
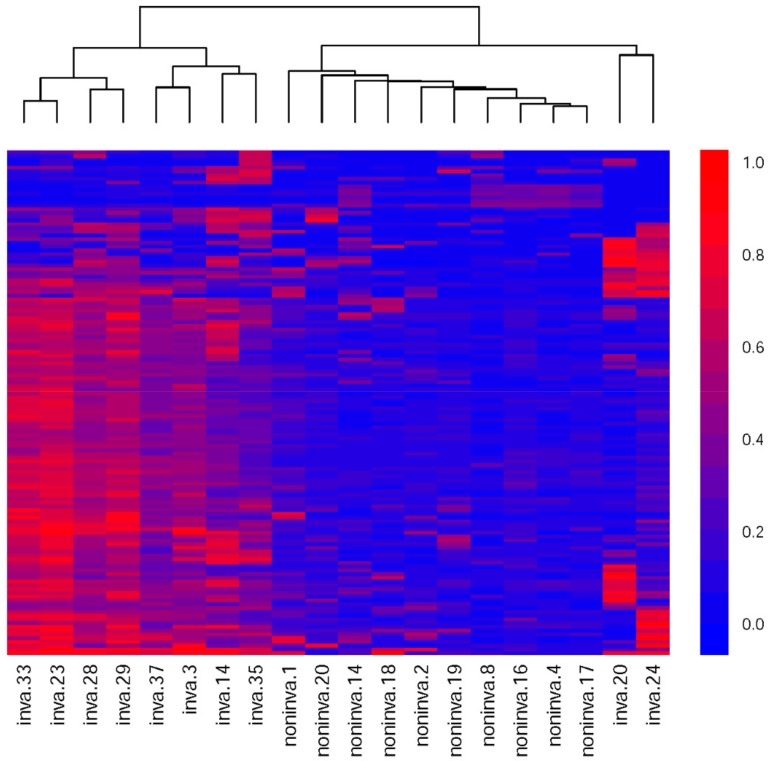
Unsupervised clustering and heatmap of 20 samples based on the 134 differently methylated probes. Each column corresponds to a sample and each row corresponds to a CpG site. Color indicates methylation value. With color ranging from blue to red, methylation values range from small to large. Color key is to the right.

**Table 1 cells-09-00326-t001:** The number of selected CpG sites and the performance of four classifiers for DNA methylation profiles.

		Normal	Invasiveness
**MRMD**	Number of CpG	14	134
	Training Accuracy	97%	93.6%
	Testing Accuracy	96.9%	549/217
**mRMR**	Number of CpG	12	5
	Training Accuracy	99%	100%
	Testing Accuracy	96.9%	611/165
**PCA**	Number of CpG	8	3
	Training Accuracy	99%	95%
	Testing Accuracy	91%	454/312
**FA**	Number of CpG	80	60
	Training Accuracy	99%	93.3%
	Testing Accuracy	94.8%	664/102

**Table 2 cells-09-00326-t002:** Known metastasis-associated genes and their descriptions in literatures.

Gene	Description
*ABCC5*	*ABCC5* functions as a mediator of breast cancer skeletal metastasis.
*ASCL2*	Functions as a suppressor of colorectal cancer metastasis by down-regulating the *ASCL2-CXCR4* signaling axis.
*BNIP3*	“*BNIP3* deletion can be used as a prognostic marker of tumor progression to metastasis in human triple-negative breast cancer”
*FLI1*	“This study for the first time identifies *FLI1* as a clinically and functionally important target gene of metastasis, providing a rationale for developing *FLI1* inhibitors in the treatment of breast cancer.”
*ITGA6*	“The role of *PTHrP* in breast cancer growth and metastasis may be mediated via upregulation of integrin alpha6beta4 expression and Akt activation, with consequent inactivation of *GSK*-3.”
*MPL*	“In migrating cancer stem cells isolated from primary human colorectal cancers, *CD110*(+) and *CDCP1*(+) subpopulations mediate organ-specific lung and liver metastasis.”
*NCOR2*	“Thyroid hormone receptors induce TRAIL expression, and TRAIL thus synthesized acts in concert with simultaneously synthesized Bcl-xL to promote metastasis”
*RHOB*	“*RHOB* belongs to a novel class of ““genes of recurrence”“ that have a dual role in metastasis and treatment resistance.”
*SLITRK3*	*SLITRK3* expression is a highly significant predictor of gastrointestinal stromal tumor recurrence and metastasis.
*SND1*	“*SND1* is a novel *MTDH*-interacting protein and has shown that it is a functionally and clinically significant mediator of metastasis.”
*TRPS1*	“*TRPS1* plays a crucial role in osteosarcoma angiogenesis, metastasis and clinical surgical stage.”
*WWOX*	*WWOX* is associated with tumorigenicity and metastasis of head and neck and gastric signet-ring cell carcinoma.

Gene is gene symbol; “…” This is a direct quote from the corresponding literature.

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
