# Peer review of "Computational Detection of Breast Cancer Invasiveness with DNA Methylation Biomarkers"

_cells, 2020, doi:10.3390/cells9020326_

Round 1
Reviewer 1 Report
The progress of breast cancer in multiple steps from epithelial hypertrophy to highly invasive breast carcinoma involves several complex coordinated changes in gene expression programming. The invasion and metastasis of cancer are important characteristics of cancers and the main reason of death. Recent research suggests that the methylation of prometastatic genes might play an important role in cancer progression and metastasis. The phenomenon of coexistence of regional DNA hypermethylation and silencing of tumor suppressor genes in breast cancer has been the focus of research attention for years. Whereas silencing of tumor suppressor genes by hypermethylation is mainly involved in tumor growth, activation of other genes by hypomethylation seems to be targeting cancer invasiveness and metastatic potential.
In this article, authors propose a novel method to help the identification of invasive and noninvasive breast cancer for different treatments. The proposed method utilizes genome-wide methylation distance in metastatic cancer data, implemented two differential methylation analysis methods for specific CpG sites, and then trained a classifier for prediction.
Besides, the proposed methods could be a useful tool for understanding the multifaceted involvement of the DNA methylation machinery in breast cancer.
Nevertheless, the manuscript still has a few minor problems should be addressed as the following.
(1) Is there any other cancer invasive biomarkers available could be used?
(2) In the presenting research, is there any discrimination about the difference of hypermethylation and hypomethylation?
Author Response
Question (1) Is there any other cancer invasive biomarkers available could be used?
Response 1: As far as we know, there is no other cancer invasive biomarkers available. DNA methylation is the only component of the covalent DNA structure that shows cell, parent of origin, and allele-specific identity. The pattern of methylation creates a layer of information, which confers upon a genome its specific cell-type identity. Although genomes are identical in all cells of the body, the DNA methylation pattern is different in distinct cell types.
Question (2) In the presenting research, is there any discrimination about the difference of hypermethylation and hypomethylation?
Response 2: In this study, we used three genome-wide DNA methylation profiles of breast cancer and machine learning method to predict. So, there is no particular discrimination about the difference of hypermethylation and hypomethylation, because the difference is inside the training data. The difference methylation is relating the training set of samples, and the generalization is also from the training data.
Reviewer 2 Report
Re: Computational detection of breast cancer invasiveness with DNA methylation biomarkers
This manuscript by Wang et al. presents a new computational system for predicting invasiveness of breast cancer based on DNA methylation profiles. This type of cancer study per se is interesting, which solves important cancer classification problem that can generate new insights in facilitating treatment decision-making. Although mostly using standard methodologies, this study is technical sound in tacking issues related to feature reduction and deferential methylation detection. The manuscript is clearly written.
There are a few major concerns that the authors should address to improve the manuscript.
One major concern is this study lacks of comparison and discussion of metastatic cancer prediction based on gene expression where larger sets of samples are normally involved and the performances tend to be good. In addition to improved classification performance, integration of methylation data with mRNA expression may also help understand the interplay between methylation state and gene expression in driving cancer metastasis.
Is it not clear if any preprocessing is needed when combining data from different platforms, e.g., Infinium HumanMethylation450 BeadChip and Illumina 90 450k DNAm, especially when users provide their own data for new prediction.
It is unclear how to define noninvasive samples based on the Euclidean distance calculation. In the statement “We also calculated Euclidean distances between primary breast cancer in another dataset and lymph node metastases using all probes….”, we assume it refers to the paired primary cancer and reginal metastases. But how exactly to get noninvasive cases? More explanation about the rationale is expected. Additionally, why four samples with bigger distance were considered abnormal and removed?
It is unusual to validate an invasive cancer classifier using normal and primary cancer data, especially in a quantitative manner. However, the enrichment analysis based on clinical indicator is very It is unclear why those clinical indicators mostly enriched in predicted non-invasive samples.
Since most of identified genes are annotated as cancer metastasis associated in the public databases, it raises concerns about what kind of new insights can be gained from this study. Presumably those genes, based on their gene expression instead of methylation pattern, may contribute to a better classifier for cancer invasiveness, e.g., elevated expression of ABCC5 in breast cancer.
Minor,
On the website, the prediction page doesn’t provide sufficient details about what type of analysis is needed for the input data. E.g., normalization is obviously important in this case.
Standard methodologies such as PCA and factor analysis, and Hypergeometric test, shouldn’t be described in details, each taking one paragraph in methodology. Please cite the proper references and only provide synopsis as needed.
Author Response
Question 1: One major concern is this study lacks of comparison and discussion of metastatic cancer prediction based on gene expression where larger sets of samples are normally involved and the performances tend to be good. In addition to improved classification performance, integration of methylation data with mRNA expression may also help understand the interplay between methylation state and gene expression in driving cancer metastasis.
Response 1: The expression may have larger sample set and more important influence to cancers. But the expression may be regulated by multiple elements such as DNA methylation, CNV and so on. When we use expression to predict cancer metastasis, we should consider more and more elements beside DNA methylation in order to avoid the influence of other elements. To decrease the workload and redundancy, we decide to identify DNA methylation biomarkers to estimate the cancer metastasis and our work also demonstrate the reliability for predicting.
Question 2: Is it not clear if any preprocessing is needed when combining data from different platforms, e.g., Infinium HumanMethylation450 BeadChip and Illumina 90 450k DNAm, especially when users provide their own data for new prediction.
Response 2: We are so sorry to mislead you by our writing. The Infinium HumanMethylation450 BeadChip and Illumina 90 450k DNAm are the different description for one platform Illumina 450K, and we don’t need to preprocess the data. We will rewrite the content.
Question 3: It is unclear how to define noninvasive samples based on the Euclidean distance calculation. In the statement “We also calculated Euclidean distances between primary breast cancer in another dataset and lymph node metastases using all probes….”, we assume it refers to the paired primary cancer and reginal metastases. But how exactly to get noninvasive cases? More explanation about the rationale is expected. Additionally, why four samples with bigger distance were considered abnormal and removed?
Response 3: The noninvasive sample is defined by us. The Euclidean distance between cancer sample and unpaired lymph node metastases sample is larger than cancer sample and paired lymph node metastases sample, it means that the relationship between cancer sample and paired lymph node metastases sample is not significant compared with random sample pair. The lymph node metastases sample may not be caused by cancer sample. Therefore it is defined as noninvasive sample. The method of quartile can test the distribution of value in one sample. If the variable coefficient in one sample is vary large, it means that the sample is abnormal which may be caused by experimental material, laboratory contamination and many other human factors.
Question 4: It is unusual to validate an invasive cancer classifier using normal and primary cancer data, especially in a quantitative manner. However, the enrichment analysis based on clinical indicator is very It is unclear why those clinical indicators mostly enriched in predicted non-invasive samples.
Response 4:Before applying the classifier to the cancer samples, the biomarkers should have the capacity to distinguish cancer and normal samples. Therefore we use normal samples to estimate our classifier. Some primary cancer sample has the potency to infiltrate other tissues, so we decide to use primary to detect the potency and to confirm the treatment method. We are so sorry about misleading you by our writing. The Y axis represents the enrichment ratio of the clinical indicators, which is calculated by the log value of the ratio equal to dividing the proportion of clinical indicators in the subclasses by the proportion of clinical indicators in the background. Therefore the bar below the X axis means the low enrichment ratio.
Question 5: Since most of identified genes are annotated as cancer metastasis associated in the public databases, it raises concerns about what kind of new insights can be gained from this study. Presumably those genes, based on their gene expression instead of methylation pattern, may contribute to a better classifier for cancer invasiveness, e.g., elevated expression of ABCC5 in breast cancer.
Response 5: Although these genes are associated with cancers, but the combination may not have the capacity to distinguish invasiveness and noninvasiveness. The biomarkers our method filters have the capacity to forecast them. Therefore this is the new insight our method offers.
Question 6: On the website, the prediction page doesn’t provide sufficient details about what type of analysis is needed for the input data. E.g., normalization is obviously important in this case.
Response 6:The format of data is the common format like the data TCGA provides. The value is usually equal to Methy/(Methy+Unmethy) and don’t need to special normalization.
Question 7: Standard methodologies such as PCA and factor analysis, and Hypergeometric test, shouldn’t be described in details, each taking one paragraph in methodology. Please cite the proper references and only provide synopsis as needed.
Response 7: Our paper and method should provide help for different scientists. But many of them such as clinicians and biologists don’t have the math and bioinformatic foundation, so we write the content in detail to help them understand our paper and use our webserver. At last we will add the reference in our paper.
Reviewer 3 Report
The manuscript is well written . The topic of the article sounds very interesting.
I recommend for the publication.
Author Response
Response: Thank you for your review and comments.
Reviewer 4 Report
In the manuscript Wang et al suggest a method to predict breast cancer invasiveness.
However, some of the step taken are not clear and some of the results might not support the claims.
Questions:
Methods:
1) Page 3 line 98: “We calculated Euclidean distances between any two matched samples using all 393,806 probes, and then removed abnormal sample pairs with excessive distances between them using the method of quartile. Subsequently, we obtained 40 matched sample pairs whose primary breast cancer was defined as invasive. We also calculated Euclidean distances between primary breast cancer in another dataset and lymph node metastases using all probes. If the minimum distance between a sample and the 40 lymph node metastases was 10 units larger than the maximum distance of the paired sample, we defined the sample as noninvasive. In this way, we identified 20 noninvasive samples. Invasive and noninvasive samples, together with 40 normal samples, collectively constituted the training sets. ”
Not clear what the approach and data sets used in each comparison. Maybe a figure with number of sample and sample types what kind of comparison. From other parts of the manuscript I assume two data sets 44 primary and Mets plus 80 primary and normal for the training set? How did the authors control for tissue differences(breast vs lymph nodes), tumour content (low tumour content will be more similar to normal sample) and tumour heterogeneity (met can be quite different from primary as met came from a different sub-clone) in the samples? How did they control for breast cancer subtypes and Hormone positive tumours vs triple negative tumour? Did they define invasive only based on distance of the methylation levels?
2) Page 3 line 109:
“Robust EB improves differential expression analysis by strengthening the hyperparameter estimation procedure and achieves robustness with regard to inaccurate working priors by conditioning on the rank of each estimated log-fold change rather than on the actual observation. It uses log-expression values to fit linear models for each CpG site,..”
The text above does not make sense…not sure what expression has to do with methylation values?
Is this just copied from somewhere?
3) Page 4 line 166:
“Heatmaps were used to display the difference of DNA methylation levels between two groups. “
Not clear which groups? Only heat map if invasive and noninvasive… ???
4) Page4 line 170:
“After differential methylation analysis and dimensionality reduction, we generated a list of 134 variably methylated CpG sites between the invasive and noninvasive groups …”
The author have applied several methods to look for differences. But not clear how the invasive and noninvasive groups were determined they need to clarify how the groups are established. There are a lot of other signal in the methylation data, which are result of changes potentially due different cell types. Not necessary between invasive and noninvasive tumours.
Results:
5) Page 5 line 186:
“For the results of these two methods, we set thresholds of logFC ≥ 1.5 and ?-value< 0.01 to select differentially methylated sites.”
What the logFC>=1.5 stands for biologically? Normally methylation studies used Beta-value diff of 0.20 or bigger. Also did they not correct for multiple tests? Should be a q value or an adjust p-value for the cut?
Page 6 line 223:
“However, for the prediction of invasiveness, the four dimensionality reduction methods provided markedly different results. “
Is this because you have different subtypes of breast and different tumour contents? See comment question 1. Unfortunately, I’m not convinced about how the setting of the invasive and non-invasive groups was done. So all the downstream analysis will suffer.
Figure 2: not clear what is presented here is the training set or the test (TCGA) set or all them together? This graph just suggests to me that the CpG sites selected are not differentiating invasive from non-invasive and that is a problem right?
Author Response
Question 1: Page 3 line 98: “We calculated Euclidean distances between any two matched samples using all 393,806 probes, and then removed abnormal sample pairs with excessive distances between them using the method of quartile. Subsequently, we obtained 40 matched sample pairs whose primary breast cancer was defined as invasive. We also calculated Euclidean distances between primary breast cancer in another dataset and lymph node metastases using all probes. If the minimum distance between a sample and the 40 lymph node metastases was 10 units larger than the maximum distance of the paired sample, we defined the sample as noninvasive. In this way, we identified 20 noninvasive samples. Invasive and noninvasive samples, together with 40 normal samples, collectively constituted the training sets. ”
Not clear what the approach and data sets used in each comparison. Maybe a figure with number of sample and sample types what kind of comparison. From other parts of the manuscript I assume two data sets 44 primary and Mets plus 80 primary and normal for the training set? How did the authors control for tissue differences(breast vs lymph nodes), tumour content (low tumour content will be more similar to normal sample) and tumour heterogeneity (met can be quite different from primary as met came from a different sub-clone) in the samples? How did they control for breast cancer subtypes and Hormone positive tumours vs triple negative tumour? Did they define invasive only based on distance of the methylation levels?
Response 1: We will rewrite the section of data set to introduce the data in detail and add a table about the data set. The first data set with 44 primary and Mets was from GSE59000 and the second data set with 80 primary and 40 normal was from GSE66695. The information of invasive breast cancer (GSE59000) was obtained from table S1 of the reference paper (Remodeling of the Methylation Landscape in Breast Cancer Metastasis) and not defined by us. And the noninvasive was defined through the Euclidean distance by us. We don’t need to consider the tissue difference between breast and lymph nodes. Because if the Euclidean distance between primary and met is much larger, it means the met may not be caused by breast cancer and the breast cancer should be defined as noninvasive. Therefore we use 20 cancer samples and 40 normal samples from GSE66695 and 40 invasive from GSE59000 as the training set. We don’t consider the breast cancer subtypes and Hormone positive tumours vs triple negative tumour.
Question 2: Page 3 line 109: “Robust EB improves differential expression analysis by strengthening the hyperparameter estimation procedure and achieves robustness with regard to inaccurate working priors by conditioning on the rank of each estimated log-fold change rather than on the actual observation. It uses log-expression values to fit linear models for each CpG site,..” The text above does not make sense…not sure what expression has to do with methylation values? Is this just copied from somewhere?
Response 2: We are so sorry about the writing. The word “methylation” was writen as “expression” by mistake and we will correct it.
Question 3: Page 4 line 166: “Heatmaps were used to display the difference of DNA methylation levels between two groups. “Not clear which groups? Only heat map if invasive and noninvasive… ???
Response 3: The two groups are invasive and noninvasive. We will rewrite the section of heatmaps.
Question 4: Page4 line 170: “After differential methylation analysis and dimensionality reduction, we generated a list of 134 variably methylated CpG sites between the invasive and noninvasive groups …” The author have applied several methods to look for differences. But not clear how the invasive and noninvasive groups were determined they need to clarify how the groups are established. There are a lot of other signal in the methylation data, which are result of changes potentially due different cell types. Not necessary between invasive and noninvasive tumours.
Response 4: The definition of invasive and noninvasive has been written in the section “2.2 Methods: Study design”. The information of invasive breast cancer (GSE59000) was obtained from table S1 of the reference paper (Remodeling of the Methylation Landscape in Breast Cancer Metastasis) and not defined by us. And the noninvasive was defined through the Euclidean distance by us.
Question 5: Page 5 line 186: “For the results of these two methods, we set thresholds of logFC ≥ 1.5 and ?-value< 0.01 to select differentially methylated sites.” What the logFC>=1.5 stands for biologically? Normally methylation studies used Beta-value diff of 0.20 or bigger. Also did they not correct for multiple tests? Should be a q value or an adjust p-value for the cut?
Response 5: The FC means fold change. Besides difference and p-value, fold change is also an indicator to estimate the difference of DNA methylation. We has find reference to use fold change to identify the differential methylation sites (Alterations of DNA methylation profile in proximal jejunum potentially contribute to the beneficial effects of gastric bypass in a diabetic rat model).
Question 6: Page 6 line 223: “However, for the prediction of invasiveness, the four dimensionality reduction methods provided markedly different results. “ Is this because you have different subtypes of breast and different tumour contents? See comment question 1. Unfortunately, I’m not convinced about how the setting of the invasive and non-invasive groups was done. So all the downstream analysis will suffer.
Response 6: We don’t consider the subtypes and we think the subtypes don’t influence our result. Because the data set has mixed different subtypes averagely. We think the difference among different methods may be influenced by the data mass, range and distribution of data. Therefore we use different methods to find the optimal method to analyze our data, and the result of analysis also demonstrate the correctness of our method.
Question 7: Figure 2: not clear what is presented here is the training set or the test (TCGA) set or all them together? This graph just suggests to me that the CpG sites selected are not differentiating invasive from non-invasive and that is a problem right?
Response 7:The samples of data set from TCGA were all test set. The section of “3.3 TCGA breast cancer cohort confirms the performance of classifiers” has showed that they are the test set. The graph is the result of enrichment analysis of four methods. The lefttop (the method MRMD) of the graph has demonstrate the significant relationship with clinical indicators and the other three methods are not significant, which consistent with our results and show the correctness of the MRMD method we select.